# Hydrogen Sulfide Enhances Browning Repression and Quality Maintenance in Fresh-Cut Peaches via Modulating Phenolic and Amino Acids Metabolisms

**DOI:** 10.3390/foods12061158

**Published:** 2023-03-09

**Authors:** Li Wang, Chen Zhang, Kaili Shi, Shouchao Chen, Jiawei Shao, Xingli Huang, Mingliang Wang, Yanyan Wang, Qingyuan Song

**Affiliations:** Anhui Agricultural Products Processing Engineering Laboratory, Key Laboratory of Agricultural Product Fine Processing and Resource Utilization, Ministry of Agriculture and Rural Affairs, College of Tea and Food Science and Technology, Anhui Agricultural University, Hefei 210036, China; 21721018@stu.ahau.edu.cn (C.Z.); 21720097@stu.ahau.edu.cn (K.S.); carmacabrera7195@gmail.com (S.C.); 17856178804@139.com (J.S.); 21721051@stu.ahau.edu.cn (X.H.); 21721080@stu.ahau.edu.cn (M.W.); wangyyan@stu.ahau.edu.cn (Y.W.); songqy1999@stu.ahau.edu.cn (Q.S.)

**Keywords:** fresh-cut, peach, H_2_S, browning, phenolic metabolism, amino acids metabolism

## Abstract

Effects of hydrogen sulfide (H_2_S) on the browning and quality maintenance of fresh-cut peach fruit were studied. The results showed that H_2_S treatment repressed the development of surface browning, suppressed the increase in respiration rate and weight loss, and delayed the decline of firmness while soluble solids content (SSC) and microbial growth were unaffected during storage. H_2_S treatment maintained higher contents of phenolic compounds, especially neo-chlorogenic acid, catechin, and quercetin, and delayed the degradation of phenolic compounds by enhancing the activities of phenolic biosynthesis-related enzymes and inhibiting the oxidative activities of polyphenol oxidase (PPO) in comparison with control. Moreover, H_2_S stimulated the accumulation of amino acids and their derivatives including proline, γ-aminobutyric acid (GABA), and polyamines (PAs) via enhancing biosynthesis and repressing degradation compared to control. These results suggested that H_2_S treatment enhanced the accumulation of phenolic, amino acids, and their derivatives by modulating phenolic and amino acids metabolisms, which contributed to the higher antioxidant activity and membrane integrity maintenance, ultimately repressing browning development and maintaining the quality. Therefore, the current study speculated that H_2_S might be a promising approach for browning inhibition and quality maintenance in fresh-cut peach fruit.

## 1. Introduction

Fresh-cut products have become an important part of the fruits and vegetables industry and family consumption owing to their convenience and freshness in recent years [1]. Fresh-cut peach fruit is extremely popular among consumers and the processing industry due to its distinct flavor, and special nutritional and functional ingredients such as phenolics, vitamins, minerals, and amino acids [2]. However, cutting as a kind of mechanical damage destroys the integrity of fruit cells and accelerates the loss of nutrients [3]. The cut caused by cutting in peach fruit is easy to brown, and certainly susceptible to microbial infections and quality deterioration, which has a negative impact on market value. At present, various physical and chemical treatments containing nitric oxide (NO), ascorbic acid, and vacuum packaging have been performed to alleviate the browning of fresh-cut peaches [4,5], but it is still necessary to explore an effective technique that can not only inhibit browning but also maintain fruit quality.

Hydrogen sulfide (H_2_S) is traditionally known as a poisonous gas, which is harmful to human health. However, 34–65 μmol kg^−1^ of H_2_S concentration are found in various types of mammalian cells and play a positive role in physiological regulation [6]. Similarly, plants can also produce H_2_S with the content of about 100 μmol kg^−1^ [7,8]. H_2_S in low concentrations acts as a gaseous regulator like carbon monoxide and NO, which can freely pass through the cell membrane and directly combine with the corresponding target cell or molecule to regulate the process of growth and development in plants [9]. Meanwhile, increasing studies have implicated that low concentrations of H_2_S play crucial roles in regulating physiological metabolic processes, retarding senescence, and resisting abiotic stresses of horticultural products [9,10]. A previous study elucidated that H_2_S treatment eliminated the excess reactive oxygen species (ROS) by promoting enhanced antioxidant enzymes activities, thus delaying the maturation, and expanding the shelf life of kiwifruit [9]. Moreover, recent research has unveiled that H_2_S treatment has positive effects on the browning suppression by regulating the antioxidant and phenolic metabolisms of fresh-cut fruits and vegetables, such as apples and lotus roots [11,12,13]. A quite low concentration of H_2_S is used in these studies, which is within the physiological concentration range of plants and the human body, suggesting that H_2_S could be a safe and effective strategy for post-harvest treatment of fruits and vegetables. However, there is limited evidence on the effect of H_2_S treatment on browning inhibition and quality maintenance of fresh-cut peaches.

As a kind of unique nutrient in horticultural crops, the function of bioactive substances in human health has attracted scientists to study how to maintain or even enhance their content via post-harvest treatments, processing techniques, and storage conditions [14]. Cutting, as an abiotic stress, has been proven to be a convenient and innovative technique to enhance bioactive substance contents in fruits and vegetables [15]. Phenolic compounds and amino acids are considered to be two important types of bioactive substances in fruits and vegetables with a high proficiency to defend against chronic diseases such as cardiovascular disease and cancer [16]. Numerous studies have shown that the accumulation of phenolic compounds induced by cutting improves total antioxidant capacity by providing hydrogen and decomposing peroxides to prevent cells from oxidative injury in fruits and vegetables such as carrots, strawberries, and pitaya fruit [17,18,19]. Huang et al. [4] reported that peaches also responded to cutting stress and promoted a higher content of total phenolic at earlier storage, whereas NO treatment enhanced the accumulation of total phenolic. Moreover, the content of phenolic compounds in fruits and vegetables is linked to the activities of phenolic anabolism enzymes, comprising L-phenylalanin ammonia-lyase (PAL), cinnamate-4-hydroxylase (C4H), and 4-coumarate: coenzyme A ligase (4CL). Prior research demonstrated that high activities of phenolic anabolism enzymes along with enhanced contents of phenolic compounds play effective roles in diminishing ROS and protecting membranes from oxidative damage in fresh-cut pitaya [17] and carrots [18].

Amino acids and their derivatives containing proline, GABA, and PAs, are not only bioactive compounds, but also important substrate for abiotic stresses resistance [20]. Proline, as a protein-genic secondary amino acid, could protect proteins and eliminate ROS, which helps to stabilize membrane structure and keeps cells from oxidative stress [21]. Liu et al. [22] demonstrated that proline-treated potatoes induced the higher concentration of intrinsic proline and antioxidant activity, contributing to the suppression of browning occurrence. GABA, a non-protein amino acid, is a vital signal molecule to regulate the adverse stresses of post-harvest storage [23]. In carrots, a large amount of GABA was found in shredded tissue, which indicated that GABA responded to cutting stress [24]. Meanwhile, Gao et al. [25] reported that GABA treatment had a beneficial effect on browning suppression in fresh-cut potatoes, indicating that GABA was involved in browning development. PAs, comprising putrescine (Put), spermidine (Spd), and spermine (Spm), play a positive role in regulating cellular osmolarity due to their polycationic nature in physiological pH values, and their accumulation contributed to the senescence resistance in fresh-cut products [26]. However, there is little to report on how H_2_S treatment regulates phenolic and amino acids metabolisms in fresh-cut peaches. This study aims to study (1) the changes of quality attributes, including firmness, SSC, weight loss, browning index (BI), and total aerobic bacterial count (TABC), (2) the changes of phenolic metabolism, including phenolic compounds and related enzyme activities, (3) the changes of amino acid metabolism, including proline, GABA, and PAs, and related enzyme activities, which could provide a new perspective into the effect of H_2_S on browning suppression and quality maintenance.

## 2. Materials and Methods

### 2.1. Fruit Material and Treatment

Peach fruit (*Prunus persica* Batsch cv. ‘Jinhuangjin’) used for the experiment were harvested at a maturity of 57 N firmness and 16% SSC in Hefei, Anhui, China. All work surfaces and knives were wiped by 75% ethanol solution. Two hundred and sixteen fruits with similar shapes, uniform sizes, and without damage were chosen and used for three replicates, then disinfected with 200 µL L^−1^ sodium hypochlorite for 2 min for further treatment. The air-dried peaches were cut into uniform cubes (1 × 1 × 1 cm) and randomly divided into the following two groups: (i) one group was fumigated with H_2_S gas released by 1.6 mmol L^−1^ sodium hydrosulfide (NaHS) solution in 85 L closed containers for 24 h, while (ii) the other group was fumigated with purified water in 85 L closed containers for 24 h as control. After fumigation, the fruits were placed in the colorless transparent plastic boxes (170 mm × 120 mm × 60 mm) and stored at 4 ± 1 °C for 12 h. The time when the control and H_2_S treatment ended was defined as 0 h. Mesocarp samples were collected every three hours for the determination of quality attribute indexes, including firmness, SSC, weight loss, respiration rate, BI, and TABC, and frozen in liquid nitrogen, then stored at −20 °C for biochemical analysis.

### 2.2. Quality Attributes

Firmness was evaluated by a firmness tester (GY-4-J, TOP Instruments Co., Ltd., Hangzhou, China) with an 8 mm diameter probe, and the result was expressed as N.

SSC was determined with a portable refractometer (PAL-1, ATAGO, Tokyo, Japan) and the result was presented as %.

Weight loss was expressed by evaluating the weight of fruits at each sampling point, and the following formula was used to calculate: [(M_0_ − M_1_)/M_0_] × 100%, where M_0_ and M_1_ represented the initial fruit weight and the fruit weight at each sampling point, respectively.

The respiration rate was determined by a fruit and vegetable breathing tester 3051H (Technology-Research Institute, Nanjing, China), and the result was expressed as mg kg^−1^ h^−1^.

A chroma meter (CR-400, Konica Minolta Sensing, Inc, Tokyo, Japan) was used to assess the lightness (L*), and the chromaticity of green (a*) and yellow (b*) on two opposite sides of fresh-cut peach cubes. Browning index (BI) was calculated by following the formula: [100 × (p − 0.31)]/0.172, where p = (a* + 1.75 × L*)/(5.645 × L* + a* − 3.012 × b*), BI increase = [(BIp − BIo)/BIo] × 100%, where BIp and BIo represented BI on day p and zero, respectively [27].

TABC was assayed by the method of Adiani et al. [28].

### 2.3. Contents of Total Phenolic, Total Flavonoid, and Individual Phenolic Compounds

The contents of total phenolic and total flavonoid were measured with the method described by Wang et al. [29]. Folin-Colorimetric reagent was applied to measure total phenolic content, and the absorbance was recorded at 765 nm. 5 mL of methanol containing 1% HCl were used to extract the flavonoid from two grams of mesocarp tissue. 1 mL of the supernatant was blended with 1 mL of AlCl_3_ (3%) and 0.5 mL of ethanol (30%) for 20 min, and the absorbance was recorded at 430 nm. The contents of total phenolic and total flavonoid were presented as g kg^−1^ based on the fresh weight (FW) of gallic acid equivalent and rutin equivalent, respectively.

Individual phenolic compounds were measured according to the description of Wang et al. [30] with high performance liquid chromatography (HPLC, Waters 2695, Milford, MA, USA) to identify and quantify by analyzing the retention times and standard curve, respectively, and the results were represented as mg kg^−1^ FW.

### 2.4. Activities of Key Enzymes Related to Phenolic Metabolism

The activities of PAL, C4H, and 4CL were assayed using the description of Wang et al. [29], while PPO activity was examined following the method of Liu et al. [22]. One unit of these enzyme activities was described as the enzyme capacity causing a 0.01 absorbance variation at 290, 340, 333, and 420 nm per min, respectively, and results were represented as U kg^−1^ FW.

### 2.5. GABA Content and Glutamate Decarboxylase (GAD) Activity

Two grams of peach tissue were ground with 5 mL of 50 mmol L^−1^ lanthanum chloride and centrifuged at 12,000× *g* for 10 min, then 2 mol L^−1^ KOH was added to the supernatant. After re-centrifugation, the supernatant was prepared for GABA content determination according to the description of Wang et al. [31]. The absorbance was measured at 645 nm, and the result was assessed using the GABA standard curve and represented as g kg^−1^ FW.

GAD activity was estimated as GABA formation following the method of Wang et al. [31] from two grams of peach tissue. One unit of GAD activity was described as the enzyme capacity that generated 1 g of GABA per min. The result was represented as U kg^−1^ FW.

### 2.6. Proline Content and Related Metabolic Enzymes Activities

The content of proline and the activities of proline-5-carboxylate synthetase (P5CS), ornithine δ-amino-transferase (OAT), and proline dehydrogenase (PDH) were measured from two grams of tissue sample based on the method of Wang et al. [30]. Proline content was assayed with the calibration curve and represented as g kg^−1^ FW. The enzyme capacity causing a 0.01 absorbance variation at 340 nm per min was described as one unit of P5CS and PDH activity. One unit of OAT activity was described as the enzyme capacity of generating 1 mol P5C per min at 510 nm. The activities of these enzymes were represented as U kg^−1^ FW.

### 2.7. PAs Contents

The identify and quantify of PAs was referred to the modified method of Wang et al. [30] by using HPLC with a UV detector at 254 nm from one gram of peach tissue. Samples were ground with 3 mL of perchloric acid (5%) for 1 h. After centrifugation, 2 mL of 2 mmol L ^−1^ NaOH, 10 μL benzoyl chloride, and 2 mL of supernatant composed of the reaction system, which was then reacted for 30 min at 37 °C. Then 3 mL of saturated NaCl and 2.5 mL of precooled ether were blended with the reaction mixture. After re-centrifugation, the ether phase was dried and diluted in 0.5 mL of methanol, then passed through a 0.22 μm filter for PAs measurement. The mobile phase was methanol at a concentration of 65.5% (*v*/*v*), injected sample volume was 20 μL, and the column operating temperature and flow rate were maintained at 30 °C and 0.8 mL min^−1^, respectively. PAs quantification was based on a calibration curve and the result was presented as mol kg^−1^ FW.

### 2.8. Activities of Key Enzymes Related to PAs Metabolism

Arginase activity was assayed based on the modified description of Bokhary et al. [32], and one unit of arginine activity was described as the capacity of 1 mmol urea formation per min. The activities of ADC, ODC, PAO, and DAO were determined following the method of Wang et al. [26], while one unit of ADC or ODC activity was described as the capacity of 1 mol of Agm or Put production per hour at 254 nm, and one unit of PAO or DAO activity was described as the enzyme capacity causing a 0.01 absorbance variation at 550 nm per min. Results of enzyme activities were presented as U kg^−1^ FW.

### 2.9. Data Analysis

All experiments were assayed with a completely random design and measured three replicates. Results were represented as mean ± standard error. The difference between treatments was analyzed using a one-way analysis of variance (ANOVA) and SPSS (version 9.1, Chicago, IL, USA), and a *p*-value less than 0.05 was considered as significant.

## 3. Results

### 3.1. Firmness, SSC, Weight Loss, Respiration Rate, BI, and TABC of Fresh-Cut Peaches

As shown in Table 1, the firmness showed a slightly slow downward trend during storage time while H_2_S treatment maintained a slightly higher firmness of fresh-cut peaches in the middle stage of storage, but there was no significant difference between the control and H_2_S treatments. SSC showed a similar tendency with a small range of change in control and H_2_S treatment during storage time. The respiration rate of fresh-cut peaches rose considerably during storage. And the respiration rate of fresh-cut peaches in control was always higher than that in H_2_S treatment. The weight loss rate of fresh-cut peaches increased dramatically within 3 h of storage and then slowed down due to the increased contact area with air during cutting. Compared with the control, the weight loss rate of H_2_S treatment was lower as a whole, and the growth range of weight loss rate was also smaller. A steady rise in BI of fresh-cut peaches was discovered throughout the storage period, and the significant lower BI was found in H_2_S treatment (*p* < 0.05). Moreover, H_2_S treatment showed a lower TABC compared to control in fresh-cut peaches during storage, but there was no significant difference with the control.

### 3.2. Contents of Total Phenolic, Total Flavonoid, and Individual Phenolic Compounds of Fresh-Cut Peaches

Total phenolic and total flavonoid contents in both control and H_2_S treatments exhibited an increasing tendency during storage (Figure 1). H_2_S treatment induced the accumulation of total phenolic and total flavonoid in fresh-cut peaches and maintained them at the higher levels in comparison with control during storage. After 12 h of storage, the contents of total phenolic and total flavonoid in H_2_S-treated fresh-cut peaches were 3.0% and 8.7% higher than those in control, respectively (Figure 1A,B).

The changes of individual phenolic compounds were displayed in Table 2. Eight phenolic compounds contained cyanidin-3-glucoside, catechin, chlorogenic/neo-chlorogenic acid, quercetin-3-rutinoside/glucoside/galactoside, and kaempferol-3-rutinoside were identified and quantified in ‘Jinhuangjin’ fresh-cut peaches, with neo-chlorogenic acid being the most abundant followed by chlorogenic acid. The contents of these compounds in both treatments showed a trend of first increasing and then decreasing, in which neo-chlorogenic acid reached the peak value at 9 h, while other individual phenolic compounds peaked at 6 h. The contents of neo-chlorogenic acid, cyanidin-3-glucoside, catechin, and chlorogenic acid in H_2_S-treated fresh-cut peaches ranged from 48.37–61.59, 0.77–1.95, 3.14–5.56, and 15.56–21.10 mg kg^−1^ during storage, respectively. The increase of quercetin-3-rutinoside/glucoside/galactoside and kaempferol-3-rutinoside was remarkably promoted by H_2_S treatment in comparison with control.

### 3.3. Activities of Key Enzymes Related to Phenolic Metabolism

PAL activity of fresh-cut peaches increased continuously during the whole storage time, and the dynamic change of PAL activity was similar to that of total phenolic content. Meanwhile, PAL activity was significantly (*p* < 0.05) different between control and treatment in fresh-cut peaches (Figure 2A). Moreover, C4H and 4CL activities increased gradually and then decreased after 9 h (Figure 2B,C). H_2_S treatment promoted the higher activities of 4CL and C4H, which brought a 5.1% and 16.7% increase in 4CL and C4H activity at 9 h in comparison with the control, respectively. PPO activity dramatically declined at 3 h and then rose until the end of storage in control, while it continued to decline in H_2_S treatment. PPO activity in H_2_S treatment was 66.7% lower than that of the control after 12 h of storage (Figure 2D).

### 3.4. GABA Content and GAD Activity

GABA content of fresh-cut peaches in both treatments rose rapidly with a storage period (Figure 3A). H_2_S treatment induced and maintained a higher level of GABA content compared to the control. At the end of storage, GABA content in H_2_S treatment increased by 10.7% compared with the control. GAD activity increased gradually and decreased thereafter, which peaked at 9 h (Figure 3B). H_2_S treatment maintained higher GAD activity, with increases of 13.5% and 11.9% compared to control at 3 h and 9 h, respectively.

### 3.5. Proline Content and Related Metabolic Enzymes Activities

Proline content generally increased first and decreased afterwards, which peaked at 3 h in H_2_S treatment and 6 h in the control, respectively (Figure 4A). H_2_S treatment induced the proline accumulation and retained a higher level than control during storage. The level of proline in H_2_S-treated fresh-cut peaches was three times higher than in control at 3 h.

The change of P5CS activity was consistent with that of proline content. H_2_S treatment induced higher levels of P5CS activity than control during 3 h of storage (Figure 4B). OAT activity in H_2_S treatment peaked at 9 h, whereas OAT activity in control increased steadily (Figure 4C). PDH activity increased in control while decreasing in H_2_S treatment at 3 h, then gradually increased in both treatments. H_2_S treatment up-regulated OAT activity but inhibited PDH activity, which promoted proline accumulation (Figure 4C,D).

### 3.6. PAs Contents and Related Key Enzymes Activities

Put was the main amine in fresh-cut peaches, followed by Spd and Spm (Figure 5A–C). The contents of Put and Spm increased first and then decreased during storage, which reached the peak at 9 h and 6 h, respectively. The contents of Put and Spm in H_2_S treatment were significantly (*p* < 0.05) higher than those in control (Figure 5A,C). The Spd content of the two treatments exhibited an increasing trend, and the Spd content of H_2_S treatment was always significantly (*p* < 0.05) higher than that of control (Figure 5B). After 12 h of storage, the Put and Spd contents of fresh-cut peaches treated with H_2_S were 19.8% and 6.7% higher compared with control, respectively. Therefore, H_2_S treatment may promote the accumulation of amines in fresh-cut peaches.

The activities of arginase, ADC, ODC, PAO, and DAO of fresh-cut peach in H_2_S treatment and control showed a trend of first increased and then decreased (Figure 5D–H). Arginase activity in H_2_S-treated peaches was 14.4% higher than control during 3 h of storage (Figure 5D). The activity of ADC and ODC enzymes peaked at 9 h except for DAO and PAO activities, which reached the peak at 6 h after storage. Compared with the control, H_2_S treatment significantly (*p* < 0.05) enhanced the activities of ADC and ODC, while inhibiting PAO and DAO activities (Figure 5E–H).

## 4. Discussion

H_2_S, as the third primary gaseous transmitter, has been confirmed to be a positive tactic on inhibiting the browning of lotus roots, Chinese water chestnuts, and apples during the fresh-cut procedure [11,12,13]. However, few studies have been conducted on its physiological functions in regulating browning and quality deterioration, the main commercial and industrial problems of fresh-cut peach fruit. Therefore, the current study investigated some quality and physiological indexes to explore the effect of H_2_S treatment on fresh-cut peach fruit, which could provide valuable data and insights for the endogenous regulation of H_2_S in post-harvest physiology of horticultural products. Browning and quality deterioration are the principal limited factors on quality maintenance of fresh-cut apples, which have a strong and negative impact on the purchase desire of consumers [13]. In this study, BI, the intensity of the brown color, increased continuously throughout the storage time, whereas the increase of BI in H_2_S-treated fruit was significantly suppressed. Firmness, SSC, weight loss, respiration rate, and TABC were used to appraise the effect of H_2_S on the sensory quality and security of fresh-cut peaches, respectively. Results illustrated that the changes of firmness and SSC were relatively stable, which may be related to the shorter storage time. H_2_S treatment maintained slightly higher firmness and SSC from 3 to 9 h of storage. The lower weight loss in H_2_S treatment was attributed to more water retention, which led to higher turgor pressure in cells; thereby contributing to the maintenance of firmness. Respiration rate in fresh-cut peaches increased immediately in both control and H_2_S treatments, which was related to physical injury. Whereas, H_2_S treatment inhibited the rise of respiration rate, leading to lower weight loss, ultimately contributing to the quality maintenance of fresh-cut peaches. TABC was a significant factor in determining the shelf life of fresh-cut fruit [23], which typically required 6 to 9 log CFU g^−1^. TABC in control and H_2_S treatments were all less than 6 log CFU g^−1^ at the end of storage, revealing the security of fruit during storage time. Moreover, Morteza et al. [8] reported that peach fruit contained about 20 μmol kg^−1^ of H_2_S, and exogenous H_2_S treatment increased the H_2_S content by about 4 times, which was still within the physiological concentration range. The concentration of H_2_S applied in the present study was also quite low and far below the safe critical concentration of 20 mg L^−1^, thus indicating that it should not had adverse effects on human health. Therefore, H_2_S might be a safe and beneficial strategy to inhibit browning and maintain the quality of fresh-cut peaches.

Phenolic compounds are commonly acknowledged to be essential antioxidants and important nutrients for human health. Current study showed that cutting-stress promoted the continuous increase of total phenolic and total flavonoid within 12 h in peaches, which coincided with previous results that significant increases in total phenolic content after cutting in pitaya and strawberries [17,19]. Meanwhile, H_2_S treatment enhanced the cutting-induced accumulation of total phenolic and total flavonoid, which were consistent with H_2_S-treated fresh-cut lotus roots and apples [11,13], unveiling that the application of H_2_S combined with cutting stress could synergistically induce the accumulation of total phenolic and total flavonoid during a shorter storage time. Moreover, Li et al. [33] reported that phenolic compounds were regarded as the crucial secondary metabolites that contributed to the antioxidant capacity in fruits. In the current study, the higher contents of total phenolic and total flavonoid in H_2_S-treated fresh-cut peaches were directly associated with the accumulation of individual phenolic profiles, including cyanidin-3-glucoside, catechin, chlorogenic/neo-chlorogenic acid, quercetin-3-rutinoside/glucoside/galactoside, and kaempferol-3-rutinoside. Among these individual phenolic compounds, H_2_S treatment remarkably enhanced the contents of neo-chlorogenic/chlorogenic acid, and maintained the most obvious neo-chlorogenic acid accumulation in fresh-cut peaches, which indicated neo-chlorogenic acid might act as the main individual phenolic compounds to provide cell antioxidant protection to resist cutting stress. This positive effect of H_2_S on improving the levels of individual phenolic compounds was also found in glycine betaine-treated intact peach fruit [29], which suggested H_2_S could induce the phenolic metabolism, thus contributing to the accumulation of phenolic compounds. Furthermore, PAL, 4CL, and C4H are vital enzymes in phenolic metabolism that collaborate to regulate the synthesis and utilization of phenolic compounds [23]. PAL is the principal enzyme that catalyzes phenylalanine into cinnamic acid. 4CL and C4H synthesize the precursors of phenolic profiles 4-hydroxycinnamic acid and p-coumaroyl CoA, respectively. Apart from the enzyme involved in phenolic compounds synthesis, PPO acts as the phenolic compound oxidase enzyme, which promotes the browning process of fruits [29]. Li et al. [17] claimed that higher activities of PAL, C4H, and 4CL were accompanied with higher levels of phenolic compounds accounting for membrane stability and enhanced cutting-stress tolerance in methyl jasmonate-treated fresh-cut pitaya fruit. A similar result was also reported in UV-C-treated strawberries [33]. In the current study, the increased activities of PAL, C4H, and 4CL were related to the accumulation of phenolic compounds and antioxidant activity in cutting-stressed peaches after 12 h of storage, while H_2_S-treated fresh-cut peaches maintained the significantly higher activities, contributing to the enhanced ROS scavenging and membrane protection. In addition, this study showed that the suppressed PPO activity along with the enhanced contents of the phenolic compounds in H_2_S treatment might play vital roles in the inhibition of surface browning in fresh-cut peaches, which was consistent with H_2_S-treated apples [13] and lotus roots [11]. Therefore, it could be postulated that H_2_S treatment has a positive effect on regulating phenolic metabolism during storage, which not only contributed to the browning inhibition, but also to the enhancement of phenolic compounds, thus improving sensory quality and nutritional value.

Accumulating studies point out that proline, GABA, and PAs are not only known as vital bioactive components, but also stress regulator molecules that play important roles in coping with browning development and adverse conditions in fruits [21,34]. Proline acts as an osmotic regulator to stabilize the structure of membranes and proteins and maintain cellular functions by scavenging hydroxyl radicals [2]. P5CS, OAT, and PDH are closely related to the accumulation or degradation of proline content, among which P5CS and OAT mediate the biosynthesis of proline via the glutamate and ornithine pathways, while PDH regulates the degradation of proline through the oxidation of proline to pyrroline-5-carboxylate, thus regulating its accumulation and metabolism [35]. Positive correlations between the accumulation of proline and the inhibition of cutting stress induced browning in fresh-cut fruits [36]. Liu et al. [22] showed that enhanced proline content contributed to the browning alleviation by regulating browning-related enzymes and substrates in proline pretreated fresh-cut potatoes. The current study showed that the accumulation of proline was associated with improved activities of P5CS and OAT and reduced PDH activity, while H_2_S treatment remarkably enhanced the biosynthesis enzymes activities and suppressed the degradation enzyme activity, which contributed to retarding the browning of fresh-cut peaches during storage time. Similar results were found in glycine betaine-treated intact peaches, the higher activities of P5CS and OAT promoted proline accumulation to induce the antioxidant capacity, resulting in increased stress resistance [37]. According to these results, H_2_S treatment-induced higher proline content might have a positive effect on eliminating ROS and restricting lipid peroxidation, thereby contributing to cellular membrane protection and quality maintenance in fresh-cut peaches during storage time.

GABA, as a signal molecule, is essential for membrane protection, sub-cellular structures stabilization, and stress regulation, and its accumulation has been frequently discovered under adverse stresses in plants [38]. Prior studies pointed out that enhanced CO_2_ content and exogenous GABA application were effective in alleviating the browning in pears [2,39]. Here, a dramatic rise in GABA content was recorded in fresh-cut peach fruit, which speculated GABA accumulation was involved in cutting stress. Meanwhile, H_2_S enhanced the beneficial effect on GABA accumulation of peaches during storage, which was in accordance with a prior study that calcium might act as signal molecular to induce the GABA accumulation under cutting stress [32], indicating that H_2_S might also play a similar role as a signal molecule. Moreover, the present study found that the accumulation of GABA was attributed to higher GAD activity, which catalyzed glutamate to GABA [40]. H_2_S treatment maintained higher GAD activity leading to the higher GABA content compared with control, which played beneficial roles in resisting cutting stress. These results were in accordance with CO_2_-treated pears, in which higher GAD activity was correlated to the accumulation of GABA, which was partly responsible for the browning repression [2]. Furthermore, prior research has demonstrated that GABA could be converted from PAs degradation by DAO and PAO under abiotic stresses [41]. In this study, H_2_S treatment suppressed the activities of DAO and PAO, speculating that H_2_S induced the accumulation of GABA was primarily synthesized through GABA shunt rather than PAs degradation pathway. Consistent with CaCl_2_-treated fresh-cut pear, Chi et al. [42] pointed out a low contribution proportion of PAs degradation pathway for GABA generation.

PAs also serve as free-radical scavengers and antioxidants to eliminate the cytosolic ROS and maintain cell redox homeostasis [43]. Accumulating evidence suggested that PAs accumulation had positive effects on abiotic stresses in fruits and vegetables [32,44]. Cao et al. [45] speculated that the alleviation of fruit softening and browning in peaches might be attributed to the accumulation of Spd and Spm induced by cold stress during early storage. Similar results were also reported in melatonin-treated peaches, in which the reduction of browning was related to up-regulated contents of PAs during cold storage [46]. In current study, Put was the principal PAs, followed by Spm in ‘Jinhuangjin’ peaches, which was consistent with ‘Yuhua’ peaches [30]. Cutting stress, as do cold stress, also generally promoted the increased contents of Put, Spd, and Spm during early storage. Meanwhile, higher PAs contents in H_2_S-treated peaches were accompanied by lower browning in comparison with control, which might be conducive to maintaining cell redox homeostasis and alleviating browning during storage. Moreover, evidence demonstrated that H_2_S treatment promoted the enhancement of PAs contents in *Trigonella foenum-graecum* under cadmium stress [47] and *Spinacia oleracea* under drought stress [48], which played a vital role in improving abiotic stress tolerance. Therefore, it could be deduced that H_2_S induced the PAs accumulation and could account for the browning mitigation, which provided the capacity to retard oxidative stress and maintain membrane integrity. In plants, PAs are generated from arginine or ornithine by ADC or ODC, respectively, and degraded by DAO and PAO [49]. The current study showed that enhanced activities of ADC and ODC and suppressed activities of PAO and DAO, which promoted PAs synthesis and inhibited PAs degradation, coincided with PAs accumulation in H_2_S treatment, thus contributing to suppressing the browning development. Similar results reported that the inhibition of melatonin on browning in peaches was associated with the higher enzyme transcription in PAs synthesis pathway and the lower enzyme transcription in PAs degradation pathway during cold storage [46]. Furthermore, it has been generally reported that the generation of PAs, especially Put, mainly depends on ADC pathway [50]. Interestingly, this study observed that the ODC pathway also participated in Put production in H_2_S-treated peaches, which was consistent with H_2_S-treated *Spinacia oleracea* seedlings [48], where H_2_S mediated Put accumulation via both the ADC and ODC pathways to resist drought stress. Thus, it was worth noting that H_2_S treatment had beneficial effect on PAs accumulation, contributing to membrane stability and signal pathway regulation. Furthermore, H_2_S treatment could be deduced to play a beneficial role in modulating amino acids metabolism, which contributed to the browning inhibition and the enhancement of proline, GABA, and PAs, thereby maintaining fruit quality during storage.

## 5. Conclusions

To sum up, the present study suggested that H_2_S treatment could be a safe and useful tactic for fresh-cut peach fruit quality maintenance and browning suppression. Meanwhile, phenolic and amino acids metabolisms might be implicated in the browning of fresh-cut peach fruit. H_2_S treatment stimulated the accumulation of phenolic compounds by enhancing the activities of phenolic biosynthesis-related enzymes (PAL, C4H, 4CL), and delayed the degradation of phenolic compounds by inhibiting the oxidative activities of PPO. Moreover, H_2_S maintained the higher levels of proline, GABA, and PAs via enhancing biosynthesis and repressing degradation. Therefore, H_2_S treatment modulated phenolic and amino acids metabolisms might contribute to the higher antioxidant activity and membrane integrity maintenance, ultimately repressing browning development and maintaining the quality. However, surface browning of fresh-cut peaches is a complex process; thus, further research about the changes of other metabolism and molecular levels are required for a better understanding.

## Figures and Tables

**Figure 1 foods-12-01158-f001:**
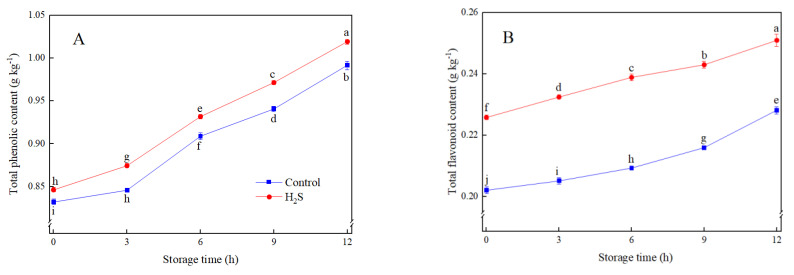
Effects of H_2_S treatment on the content of total phenolic (**A**) and total flavonoid (**B**) of fresh-cut peaches during storage. Data are presented as mean ± standard errors of triplicate samples. Different letters indicate statistically significant (*p* < 0.05) differences.

**Figure 2 foods-12-01158-f002:**
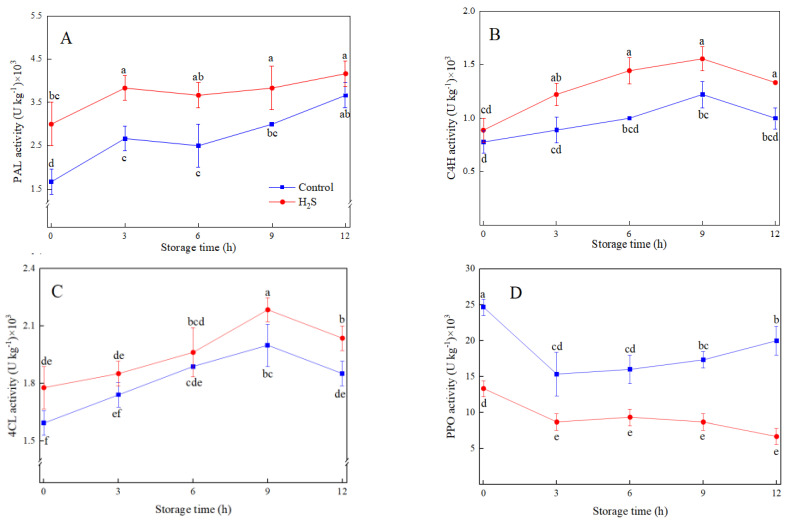
Effects of H_2_S treatment on the activities of PAL (**A**), 4CL (**B**), C4H (**C**), and PPO (**D**) of fresh-cut peaches during storage. Data are presented as mean ± standard errors of triplicate samples. Different letters indicate statistically significant (*p* < 0.05) differences.

**Figure 3 foods-12-01158-f003:**
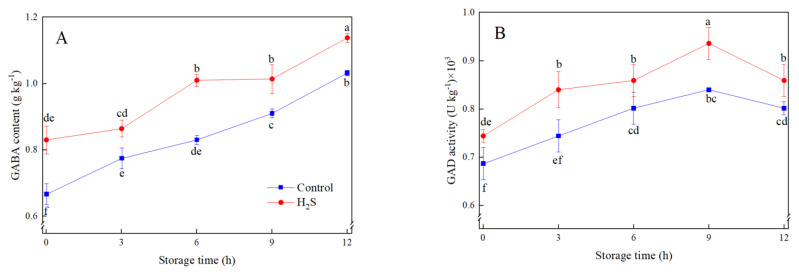
Effects of H_2_S treatment on GABA content (**A**) and GAD activity (**B**) of fresh-cut peaches during storage. Data are presented as mean ± standard errors of triplicate samples. Different letters indicate statistically significant (*p* < 0.05) differences.

**Figure 4 foods-12-01158-f004:**
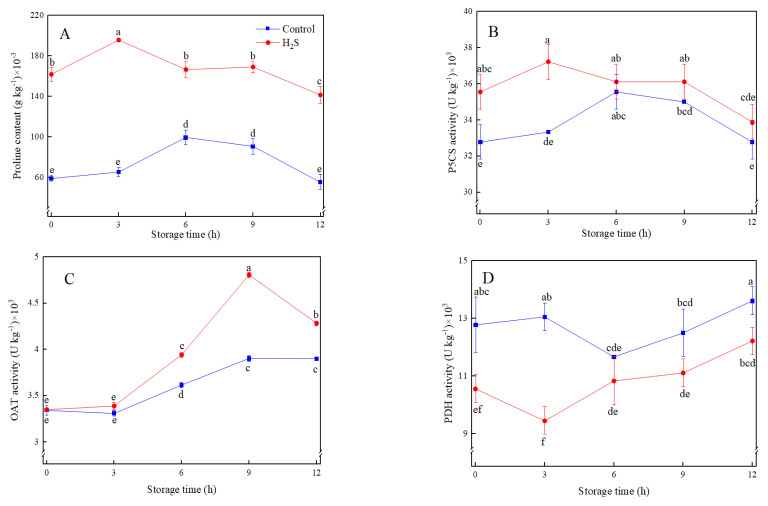
Effects of H_2_S treatment on proline content (**A**) and the activities of P5CS (**B**), OAT (**C**), and PDH (**D**) of fresh-cut peaches during storage. Data are presented as mean ± standard errors of triplicate samples. Different letters indicate statistically significant (*p* < 0.05) differences.

**Figure 5 foods-12-01158-f005:**
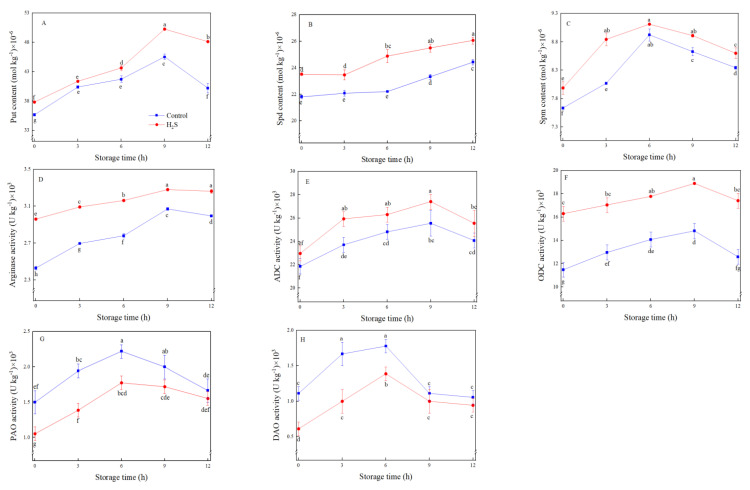
Effects of H_2_S treatment on the content of Put (**A**), Spd (**B**), and Spm (**C**), and the activities of arginase (**D**), ADC (**E**), ODC (**F**), PAO (**G**), and DAO (**H**) of fresh-cut peaches during storage. Data are presented as mean ± standard errors of triplicate samples. Different letters indicate statistically significant (*p* < 0.05) differences.

**Table 1 foods-12-01158-t001:** Effects of H_2_S treatment on weight loss, SSC, firmness, respiration rate, BI, and TABC in fresh-cut peaches during storage.

Storage Time (h)	Treatment	Weight Loss (%)	SSC (%)	Firmness (N)	Respiration Rate (mg kg^−1^ h^−1^)	BI	TABC (log CFU g^−1^)
0	Control	0	15.51 ± 0.63 ^b^	57.43 ± 0.32 ^a^	270.10 ± 5.07 ^d^	0	2.36 ± 0.18 ^a^
	H_2_S	0	16.61 ± 0.38 ^a^	57.18 ± 0.69 ^a^	242.76 ± 5.45 ^f^	0	2.05 ± 0.51 ^a^
3	Control	0.12 ± 0.07 ^c^	14.77 ± 0.40 ^c^	57.36 ± 0.29 ^a^	298.96 ± 7.31 ^b^	29.38 ± 0.17 ^d^	2.36 ± 0.93 ^a^
	H_2_S	0.12 ± 0.00 ^c^	14.78 ± 0.19 ^bc^	57.39 ± 0.28 ^a^	267.19 ± 5.71 ^d^	17.85 ± 0.06 ^g^	2.04 ± 0.96 ^a^
6	Control	0.22 ± 0.01 ^b^	15.00 ± 0.82 ^b^	56.42 ± 1.12 ^a^	305.84 ± 7.08 ^a^	34.82 ± 0.22 ^c^	2.37 ± 0.88 ^a^
	H_2_S	0.15 ± 0.00 ^c^	15.14 ± 0.14 ^b^	57.28 ± 1.30 ^a^	270.38 ± 4.95 ^d^	19.83 ± 0.33 ^f^	2.05 ± 0.80 ^a^
9	Control	0.34 ± 0.02 ^a^	15.04 ± 0.60 ^b^	56.44 ± 0.95 ^a^	292.04 ± 2.86 ^c^	40.44 ± 0.09 ^b^	2.39 ± 0.69 ^a^
	H_2_S	0.18 ± 0.03 ^bc^	16.03 ± 0.34 ^a^	57.14 ± 0.60 ^a^	244.92 ± 4.14 ^f^	27.86 ± 0.08 ^e^	2.07 ± 0.85 ^a^
12	Control	0.38 ± 0.04 ^a^	16.27 ± 0.41 ^a^	55.47 ± 0.37 ^a^	296.09 ± 5.19 ^b^	42.11 ± 0.25 ^a^	2.34 ± 1.79 ^a^
	H_2_S	0.21 ± 0.02 ^b^	16.10 ± 0.17 ^a^	55.59 ± 0.39 ^a^	259.46 ± 5.66 ^e^	33.75 ± 0.14 ^c^	2.08 ± 0.90 ^a^

Data are presented as mean ± standard errors of triplicate samples. Different letters indicate statistically significant (*p* < 0.05) differences.

**Table 2 foods-12-01158-t002:** Effects of H_2_S treatment on individual phenolic compounds in fresh-cut peaches during storage.

Storage (h)	Treatment	Cyanidin-3- Glucoside	Catechin	Chlorogenic Acid	Neo-Chlorogenic Acid	Quercetin-3-Rutinoside	Quercetin-3-Glucoside	Kaempferol-3-Rutinoside	Quercetin-3-Galactoside
0	Control	0.77 ± 0.09 ^g^	3.14 ± 0.00 ^h^	15.56 ± 0.00 ^f^	48.37 ± 0.12 ^e^	2.83 ± 0.03 ^c^	1.97 ± 0.01 ^d^	1.21 ± 0.01 ^f^	0.75 ± 0.02 ^a^
	H_2_S	1.54 ± 0.38 ^e^	4.66 ± 0.01 ^f^	19.23 ± 0.01 ^e^	54.98 ± 0.08 ^d^	2.84 ± 0.01 ^c^	2.04 ± 0.01 ^c^	1.34 ± 0.01 ^e^	0.87 ± 0.04 ^a^
3	Control	1.08 ± 0.07 ^f^	4.22 ± 0.06 ^g^	19.24 ± 0.08 ^e^	44.00 ± 0.14 ^g^	2.84 ± 0.01 ^c^	1.85 ± 0.03 ^e^	1.19 ± 0.01 ^f^	0.70 ± 0.03 ^a^
	H_2_S	1.56 ± 0.07 ^e^	4.49 ± 0.04 ^f^	22.66 ± 0.00 ^c^	45.39 ± 0.34 ^f^	2.87 ± 0.01 ^b^	2.12 ± 0.01 ^b^	1.33 ± 0.00 ^e^	0.81 ± 0.07 ^a^
6	Control	3.03 ± 0.26 ^b^	6.09 ± 0.02 ^b^	23.14 ± 0.68 ^b^	58.63 ± 0.58 ^c^	2.88 ± 0.01 ^b^	2.12 ± 0.01 ^b^	1.37 ± 0.01 ^d^	0.78 ± 0.01 ^a^
	H_2_S	3.63 ± 0.21 ^a^	6.30 ± 0.03 ^a^	23.64 ± 0.03 ^a^	58.78 ± 0.26 ^c^	2.92 ± 0.01 ^a^	2.28 ± 0.02 ^a^	1.61 ± 0.01 ^a^	0.91 ± 0.15 ^a^
9	Control	2.53 ± 0.10 ^c^	5.78 ± 0.04 ^c^	21.22 ± 0.06 ^c^	61.91 ± 0.76 ^b^	2.86 ± 0.01 ^b^	1.97 ± 0.00 ^d^	1.43 ± 0.01 ^c^	0.80 ± 0.09 ^a^
	H_2_S	2.65 ± 0.10 ^c^	6.03 ± 0.01 ^b^	21.58 ± 0.04 ^c^	62.80 ± 0.02 ^a^	2.90 ± 0.01 ^a^	2.06 ± 0.01 ^c^	1.50 ± 0.01 ^b^	0.83 ± 0.08 ^a^
12	Control	1.91 ± 0.19 ^d^	5.40 ± 0.01 ^e^	20.00 ± 0.05 ^d^	59.44 ± 0.37 ^c^	2.86 ± 0.01 ^b^	1.96 ± 0.01 ^d^	1.33 ± 0.01 ^e^	0.82 ± 0.04 ^a^
	H_2_S	1.95 ± 0.16 ^d^	5.56 ± 0.01 ^d^	21.10 ± 0.02 ^c^	61.59 ± 0.21 ^b^	2.91 ± 0.00 ^a^	2.03 ± 0.01 ^c^	1.34 ± 0.01 ^e^	0.86 ± 0.11 ^a^

Data are presented as mean ± standard errors of triplicate samples. Different letters indicate statistically significant (*p* < 0.05) differences.

## Data Availability

The data presented in this study are available on request from the corresponding author. The data are not publicly available due to privacy restrictions.

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
