# Peer review of "Hydrogen Sulfide Enhances Browning Repression and Quality Maintenance in Fresh-Cut Peaches via Modulating Phenolic and Amino Acids Metabolisms"

_foods, 2023, doi:10.3390/foods12061158_

Round 1

Reviewer 1 Report

In this paper, the authors described the effect of the hydrogen sulfide gas treatment on the cutted peach to identify the usefulness for the food preservation. The data here indicates that the preservation of nutrients in cutted peach was enhanced by hydrogen sulfide treatment. However, this paper needs some revisions for the following reasons.

1. The safety level of hydrogen sulfide using here should be discussed. In general, hydrogen sulfide is a highly toxic substance. Are these really supposed to be eaten?

2. The difference between hydrogen sulfide-treated and untreated values at 0 hour is too large among the every figures. Why not make it a relative comparison? For example, in Fig 1A, I think the change is almost the same.

Minor points

L69: NO should be abbreviated at Line 45.

L140: 'performed' should be 'measured'.

L150: 'GAD' should be fully spelled somewhere. Glutamate decarboxylase?

L163: 'performed' should be 'measured'.

Section 3.1: I think statistical 'P>0.05' value means that there is no significant difference between two data, and the words 'increased' and 'suppressed' cannot be used. 

Author Response

Reviewer #1:

Thanks for the review’s effort to improve the quality of our manuscript. We highlighted the changes in red color in our revised manuscript.

Point 1: The safety level of hydrogen sulfide using here should be discussed. In general, hydrogen sulfide is a highly toxic substance. Are these really supposed to be eaten?

Response1: Line 44-47, 58-61, 390-395: It has been added more information about safety level of H2S in the section of introduction and discussion.

Point 2: The difference between hydrogen sulfide-treated and untreated values at 0 hour is too large among the every figures. Why not make it a relative comparison? For example, in Fig 1A, I think the change is almost the same.

Response 2: The time when the control and H2S treatment ended was defined as 0 hour, which might be the reason for the large difference among figures. And this part has been added in the section of fruit material and treatment in line 118. Furthermore, the relative comparisons have been added in the figures and tables. Thank you for your constructive comments. 

Minor points

Point 1: L69: NO should be abbreviated at Line 45.

Response 1: Line 40: It has been added the abbreviation of nitric oxide.

Point 2: L140: 'performed' should be 'measured'.

Response 2: Line 150: It has been replaced “performed” by “measured”.

Point 3: L150: 'GAD' should be fully spelled somewhere. Glutamate decarboxylase?

Response 3: Line 160: It has been added fully name of GAD.

Point 4: L163: 'performed' should be 'measured'.

Response 4: Line 173: It has been replaced “performed” by “measured”.

Point 5: Section 3.1: I think statistical 'P>0.05' value means that there is no significant difference between two data, and the words 'increased' and 'suppressed' cannot be used.

Response 5: Line 208, 218: It has been corrected in section 3.1.

Reviewer 2 Report

-You should improve aim of the study

-Remove old references from introduction 

-Read all paper again and English language and style are fine/minor spell check required

-Conclusion section is short for this study and you should expand it

Author Response

Reviewer #2:

 Thanks for the review’s effort to improve the quality of our manuscript. We highlighted the changes in red color in our revised manuscript.

Point 1: You should improve aim of the study.

Response 1: Line100-103: It has been improved aim of the study.

Point 2: Remove old references from introduction.

Response 2: Line 42, 68, 139: Old references have been removed from introduction and changed by the newer references.

Point 3: Read all paper again and English language and style are fine/minor spell check required.

Response 3: Line 58, 94, 423: It has been carefully corrected in the manuscript.

Point 4: Conclusion section is short for this study and you should expand it.

Response 4: Line 523-532: It has been expanded the conclusion section.

Reviewer 3 Report

The paper “Hydrogen sulfide enhances browning repression and quality maintenance in fresh-cut peaches via modulating phenolic and amino acids metabolisms” contributes to the growth of literature for research on browning suppression and quality maintenance of fruits, especially peaches. This research contributes to the growth of literature for nutritionists, food technologists and food producers offering plant-based products. Before the manuscript acceptation for publication in “Foods” the following items should be revised:

Introduction

Are there reports on how H2S harmfulness - after consumption of thises fruits, the safety of H2S in the food chain and its interaction with other nutrients (e.g. vitamins); the antagonistic roles,  or no such influence?

Are there standards for safe consumption?

Discussion and Conclusion

What are the strength and limitations of this research?   Whether The Packaging Declaration should contain information about using this method?

Author Response

Reviewer #3:

Thanks for the review’s effort to improve the quality of our manuscript. We highlighted the changes in red color in our revised manuscript.

Point 1: Are there reports on how H2S harmfulness - after consumption of thises fruits, the safety of H2S in the food chain and its interaction with other nutrients (e.g. vitamins); the antagonistic roles, or no such influence?

Response 1: Line 44-47, 58-61, 390-395: It has been added more information about safety level of H2S in the section of introduction and discussion.

At present, few studies have shown that H2S can interact with nutrients, but H2S treatment can increase the content of nutrients by inducing the enzyme activities of related metabolic pathways. Meanwhile, H2S as a reducing agent will not react with nutrients (such as vitamins). Thank you for your constructive comments. We will further study about this part.

Point 2: Are there standards for safe consumption?

Response 2: There are no standards for safe consumption, but the dose of H2S application was quite low and within the physiological concentration, indicating that it will not have adverse effects on human health.

Point 3: What are the strength and limitations of this research?   

Response 3: Line 372-373, 530-532: It has been added more information about the strength and limitations of this research.

Point 4:  Whether The Packaging Declaration should contain information about using this method?

Response 4: We thought it is not necessary to contain information in packaging declaration. First, this method is used for preprocessing. Secondly, the dose used in this method is within the physiological concentration range of plant and human body. Some studies have shown that the concentration of fruits after treatment are also within the physiological concentration range of plant and human body, such as ‘Hydrogen sulfide prolongs postharvest shelf life of strawberry and plays an antioxidative role in fruits. J Agric Food Chem. 2012, 60, 8684-8693.’ and ‘Hydrogen sulfide treatment confers chilling tolerance in hawthorn fruit during cold storage by triggering endogenous H2S accumulation, enhancing antioxidant enzymes activity and promoting phenols accumulation. Sci Hortic. 2018, 238, 264-271.’